# Discriminative Entropy Clustering
# and its Relation to K-means and SVM

## Abstract

Maximization of mutual information between the model's input and output is formally related to "decisiveness" and "fairness" of the softmax predictions Bridle et al. (1991), motivating such unsupervised entropy-based losses for discriminative models. Recent self-labeling methods based on such losses represent the state of the art in deep clustering. First, we discuss a number of general properties of such entropy clustering methods, including their relation to K-means and unsupervised SVM-based techniques. Disproving some earlier published claims, we point out fundamental differences with K-means. On the other hand, we show similarity with SVM-based clustering allowing us to link explicit margin maximization to entropy clustering. Finally, we observe that the common form of cross-entropy is not robust to pseudo-label errors. Our new loss addresses the problem and leads to a new EM algorithm improving the state of the art on many standard benchmarks.

## 1   Introduction

Discriminative entropy-based loss functions, e.g. *decisiveness* and *fairness*, were proposed for network training Bridle et al. (1991); Krause et al. (2010) and regularization Grandvalet & Bengio (2004) and are commonly used for unsupervised and weakly-supervised classification problems Ghasedi Dizaji et al. (2017); Hu et al. (2017); Ji et al. (2019); Asano et al. (2020); Jabi et al. (2021). In particular, the state-of-the-art in unsupervised classification Asano et al. (2020); Jabi et al. (2021) is achieved by self-labeling methods using extensions of decisiveness and fairness.

Section 1.1 reviews the entropy-based clustering with soft-max models and introduces the necessary notation. Then, Section 1.2 reviews the corresponding self-labeling formulations. Section 1.3 summarizes our main contributions and outlines the structure of the main parts of the paper.

### 1.1   Discriminative entropy clustering: background and notation

Consider neural networks using probability-type outputs, e.g. *softmax* $\sigma : \mathcal{R}^K \rightarrow \Delta^K$ mapping $K$ logits $l^k \in \mathcal{R}$ to $K$-class probabilities $\sigma^k = \frac{\exp l^k}{\sum_c \exp l^c}$ forming a categorical distribution $\sigma = (\sigma^1, \dots, \sigma^K) \in \Delta^K$ often interpreted as a posterior. We reserve superscripts to indicate classes or categories. For shortness, this paper uses the same symbol for functions or mappings and examples of their output, e.g. specific predictions $\sigma$. If necessary, subscript $i$ can indicate values, e.g. prediction $\sigma_i$ or logit $l_i^k$, corresponding to any specific input example $X_i$ in the training dataset $\{X_i\}_{i=1}^N$.

The *mutual information* (MI) loss, proposed by Bridle et al. (1991) for unsupervised discriminative training of softmax models, trains the model output to keep as much information about the input as possible. They derived MI estimate as the difference between the average entropy of the output $\overline{H(\sigma)} = \frac{1}{N} \sum_i H(\sigma_i)$ and the entropy of the average output $\overline{\sigma} = \frac{1}{N} \sum_i \sigma_i$, which is a distribution of class predictions over the whole dataset

$$L_{mi} \ := \ -MI(C, X) \quad \approx \quad \overline{H(\sigma)} \ - \ H(\overline{\sigma}) \tag{1}$$

where $C$ is a random variable representing the class prediction for input $X$. Besides the motivating information-theoretic interpretation of the loss, the right-hand side in (1) has a clear discriminative interpretation that stands on its own: $H(\overline{\sigma})$ encourages "fair" predictions with a balanced support of all categories across the whole training dataset, while $\overline{H(\sigma)}$ encourages confident or "decisive" prediction at each data point suggesting that decision boundaries are away from the training examples Grandvalet & Bengio (2004). Our paper refers to unsupervised training of discriminative soft-max models using predictions' entropies, e.g. see (1), as *discriminative entropy clustering*. This should not be confused with *generative entropy clustering* methods where the entropy is used as a measure of compactness for clusters' density functions[1].

Discriminative clustering loss (1) can be applied to deep or shallow models. For clarity, this paper distinguishes parameters $\mathbf{w}$ of the *representation* layers of the network computing features $f_{\mathbf{w}}(X) \in \mathcal{R}^M$ for any input $X$. We separate the linear classifier parameters $\mathbf{v}$ in the output layer computing $K$-logit vector $l = \mathbf{v}^\top f$ for any feature $f \in \mathcal{R}^M$. As mentioned earlier, this paper uses the same notation for mapping $f(\cdot)$ and its values or (deep) features $f$ produced by the representation layers. For shortness, we assume a "homogeneous" representation of the linear classifier so that $\mathbf{v}^\top f$ includes the bias. The overall network model is defined as

$$\sigma(\mathbf{v}^\top f_{\mathbf{w}}(X)). \tag{2}$$

A special "shallow" case of the model in (2) is a basic linear discriminator

$$\sigma(\mathbf{v}^\top X) \tag{3}$$

directly operating on given input features $f(X) = X$. In this case, $M$ represents the input dimensions. Optimization of the loss (1) for the shallow model (3) is done only over linear classifier parameters $\mathbf{v}$, but the deeper network model (2) is optimized over all network parameters $[\mathbf{v}, \mathbf{w}]$. Typically, this is done via gradient descent or backpropagation Rumelhart et al. (1986); Bridle et al. (1991).

In the context of deep models (2), the decision boundaries between the clusters of data points $\{X_i\}$ can be arbitrarily complex since the network learns high-dimensional non-linear representation map or embedding $f_{\mathbf{w}}(X)$. In this case, loss (1) is optimized with respect to both representation $\mathbf{w}$ and classification $\mathbf{v}$ parameters. To avoid overly complex clustering of the training data and to improve generality, it is common to use *self-augmentation* techniques Hu et al. (2017). For example, Ji et al. (2019) maximize the mutual information between class predictions for input $X$ and its augmentation counterpart $X'$ encouraging deep features invariant to augmentation.

To reduce the model's complexity, Krause et al. (2010) combine entropy-based loss (1) with regularization of all network parameters interpreted as their isotropic Gaussian prior

$$
\begin{aligned}
L_{mi+decay} &= \overline{H(\sigma)} - H(\overline{\sigma}) + \|[\mathbf{v}, \mathbf{w}]\|^2 \\
&\stackrel{c}{=} \overline{H(\sigma)} + KL(\overline{\sigma} \,\|\, u) + \|[\mathbf{v}, \mathbf{w}]\|^2
\end{aligned} \tag{4}
$$

where $\stackrel{c}{=}$ represents equality up to an additive constant and $u$ is a uniform distribution over $K$ classes. The second loss formulation in (4) uses KL divergence motivated in Krause et al. (2010) by the possibility to generalize the fairness to any target balancing distribution different from the uniform.

### 1.2 *Self-labeling* methods for entropy clustering

Optimization of losses (1) or (4) during network training is mostly done with standard gradient descent or backpropagation Bridle et al. (1991); Krause et al. (2010); Hu et al. (2017). However, the difference between the two entropy terms implies non-convexity, which makes such losses challenging for gradient descent. This motivates alternative formulations and optimization approaches. For example, it is common to extend the loss by incorporating auxiliary or hidden variables $y$ representing *pseudo-labels* for unlabeled data points $X$, which are to be estimated jointly with optimization of the network parameters Ghasedi Dizaji et al. (2017); Asano et al. (2020); Jabi et al. (2021). Typically, such *self-labeling* approaches to unsupervised network training iterate optimization of the loss over pseudo-labels and network parameters, similarly to Lloyd's algorithm for $K$-means or EM algorithm for Gaussian mixtures Bishop (2006). While the network parameters are still optimized via gradient descent, the pseudo-labels can be optimized via more powerful algorithms.

---

[1]E.g., K-means minimizes cluster variances, whose *log*s are cluster's density entropies, assuming Gaussianity.

For example, Asano et al. (2020) formulate self-labeling using the following constrained optimization problem with discrete pseudo-labels $y$ tied to predictions by *cross entropy* function $H(y, \sigma)$

$$L_{ce} = \overline{H(y, \sigma)} \qquad s.t. \quad y \in \Delta_{0,1}^K \quad and \quad \bar{y} = u \tag{5}$$

where $\Delta_{0,1}^K$ are *one-hot* distributions, *i.e.* corners of the probability simplex $\Delta^K$. Training of the network is done by minimizing cross entropy $H(y, \sigma)$, which is convex w.r.t. $\sigma$, assuming fixed pseudo-labels $y$. Then, model predictions get fixed and cross-entropy is minimized w.r.t variables $y$. Note that cross-entropy $H(y, \sigma)$ is linear with respect to $y$, and its minimum over simplex $\Delta^K$ is achieved by one-hot distribution for a class label corresponding to $\arg\max(\sigma)$ at each training example. However, the balancing constraint $\bar{y} = u$ converts minimization of cross-entropy over all data points into a non-trivial integer programming problem that can be approximately solved via *optimal transport* Cuturi (2013). The cross-entropy in (5) encourages the network predictions $\sigma$ to approximate the estimated one-hot target distributions $y$, which implies the decisiveness.

Self-labeling methods for unsupervised clustering can also use soft pseudo-labels $y \in \Delta^K$ as target distributions inside $H(y, \sigma)$. In general, soft targets $y$ are commonly used with cross-entropy functions $H(y, \sigma)$, e.g. in the context of noisy labels Tanaka et al. (2018); Song et al. (2022). Softened targets $y$ can also assist network calibration Guo et al. (2017); Müller et al. (2019) and improve generalization by reducing over-confidence Pereyra et al. (2017). In the context of unsupervised clustering, cross entropy $H(y, \sigma)$ with soft pseudo-labels $y$ approximates the decisiveness since it encourages $\sigma \approx y$ implying $H(y, \sigma) \approx H(y) \approx H(\sigma)$ where the latter is the decisiveness term in (1). Inspired by (4), instead of the hard constraint $\bar{y} = u$ used in (5), self-labeling losses can represent the fairness using KL divergence $KL(\bar{y} \,\|\, u)$, as in Ghasedi Dizaji et al. (2017); Jabi et al. (2021). In particular, Jabi et al. (2021) formulates the following entropy-based self-labeling loss

$$L_{ce+kl} = \overline{H(y, \sigma)} + KL(\bar{y} \,\|\, u) \tag{6}$$

encouraging decisiveness and fairness, as discussed. Similarly to (5), the network parameters in loss (6) are trained by the standard cross-entropy term. Optimization over relaxed pseudo-labels $y \in \Delta^K$ is relatively easy since KL divergence is convex and cross-entropy is linear w.r.t. $y$. While there is no closed-form solution, the authors offer an efficient approximate solver for $y$. Iterating steps that estimate pseudo-labels $y$ and optimize the model parameters resemble the Lloyd's algorithm for K-means. Jabi et al. (2021) also establish a formal relation with K-means objective.

## 1.3 Summary of our contributions

Our work is closely related to self-labeling loss (6) and the corresponding ADM algorithm proposed in Jabi et al. (2021). Their inspiring approach is a good reference point for our self-labeling loss formulation (13). It also helps to illuminate the limits in a general understanding of entropy clustering.

Our paper provides conceptual and algorithmic contributions. First of all, we examine the relations of discriminative entropy clustering to K-means and SVM. In particular, we disprove the main theoretical claim (in the title) of a recent TPAMI paper Jabi et al. (2021) wrongly stating the equivalence between the standard K-means objective and the entropy-based clustering losses. Our Figure 1 provides a simple counterexample to the claim, but we also show specific technical errors in their proof. We highlight fundamental differences with a broader *generative* group of clustering methods, which includes K-means, GMM, etc. On the other hand, we find stronger similarities between entropy clustering and discriminative SVM-based clustering. In particular, this helps to formally show the soft margin maximization effect when decisiveness is combined with a norm regularization term.

This paper also proposes a new self-labeling algorithm for entropy-based clustering. In the context of relaxed pseudo-labels $y$, we observe that the standard formulation of decisiveness $\overline{H(y, \sigma)}$ is sensitive to pseudo-label uncertainty/errors. We motivate the *reverse cross-entropy* formulation, which we demonstrate is significantly more robust to label noise. We also propose a zero-avoiding form of KL-divergence as a *strong fairness* term. Unlike standard fairness, it does not tolerate highly unbalanced clusters. Our new self-labeling loss allows an efficient EM algorithm for estimating pseudo-labels. We derive closed-form E and M steps. Our new algorithm improves the state-of-the-art on many standard benchmarks for deep clustering, which empirically validates our technical insights.

Our paper is organized as follows. Section 2 discusses the relation of entropy clustering to K-means and SVM. Section 3 motivates our self-labeling loss and derives an EM algorithm for estimating pseudo-labels. The experimental results for our entropy clustering algorithm are in Section 4.

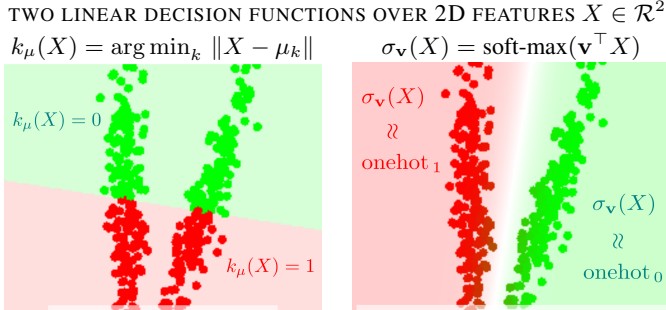

TWO LINEAR DECISION FUNCTIONS OVER 2D FEATURES $X \in \mathcal{R}^2$

$$k_\mu(X) = \arg\min_k \|X - \mu_k\| \qquad \sigma_\mathbf{v}(X) = \text{soft-max}(\mathbf{v}^\top X)$$

(a) variance clustering   (b) entropy clustering

Figure 1: K-means vs entropy clustering - binary example ($K = 2$) for 2D data $\{X_i\} \subset \mathcal{R}^M$ ($M = 2$) comparing linear methods of similar parametric complexity: (a) $K$-means [$\mu_k \in \mathcal{R}^M$] and (b) entropy clustering based on a linear classifier using $K$-columns linear discriminator matrix $\mathbf{v} = [\mathbf{v}_k \in \mathcal{R}^M]$ and soft-max predictions. Red and green colors in (a) and (b) illustrate optimal linear decision regions over $X \in \mathcal{R}^2$ produced by the decision functions $k_\mu(X)$, $\sigma_\mathbf{v}(X)$ for parameters $\mu$ and $\mathbf{v}$ minimizing two losses: (a) compactness/variance of clusters $\sum_i \|X_i - \mu_{k_i}\|^2$ where $k_i = k_\mu(X_i)$ and (b) decisiveness and fairness of predictions $\overline{H(\sigma)} - H(\bar{\sigma})$ where $H(\cdot)$ is entropy function and $\overline{H(\sigma)} = \text{avg}\{H(\sigma_i)\}$, $\bar{\sigma} = \text{avg}\{\sigma_i\}$ for $\sigma_i = \sigma_\mathbf{v}(X_i)$. The decision function $k_\mu(X)$ is hard (a) and $\sigma_\mathbf{v}(X)$ is soft, particularly near the linear decision boundary (b). The optimal results in (a,b) are analyzed in Sec.2.1. The result in (b) may require a *margin maximization* term $\|\mathbf{v}\|^2$, see Sec.2.2.

## 2  Relation to discriminative and generative clustering methods

### 2.1  Entropy-based clustering versus K-means

Discriminative entropy clustering (1) is not as widely known as K-means, but for no good reason. With linear models (3), entropy clustering (1) is as simple as K-means, e.g. it produces linear cluster boundaries. Both approaches have good approximate optimization algorithms for their non-convex (1) or NP-hard Mahajan et al. (2012) objectives. Two methods also generalize to non-linear clustering using more complex representations, e.g. learned $f_w(X)$ or implicit (kernel K-means).

There is a limited general understanding of how entropy clustering relates to more popular methods, such as K-means. The prior work, including Bridle et al. (1991), mainly discusses entropy clustering in the context of neural networks. K-means is also commonly used with deep features, but it is hard to understand the differences in such complex settings. An illustrative 2D example of entropy clustering in Krause et al. (2010) (Fig.1) is helpful, but it looks like a typical textbook example for K-means where it would work perfectly. Interestingly, Jabi et al. (2021) make a theoretical claim about algebraic equivalence between K-means objective and a regularized entropy clustering loss.

Here we show significant differences between K-means and entropy clustering. First, we disprove the claim by Jabi et al. (2021). We provide a simple *counterexample* in Figure 1 where the optimal solutions are different in a basic linear setting. Moreover, we point out a critical technical error in their Proposition 2 - its proof ignores normalization inside softmax. Symbol $\propto$ hides it in their equation (5), which is later treated as equality in the proof of Proposition 2. Equations in their proof do not work with normalization, which is critical for softmax models. The extra regularization term $\|\mathbf{v}\|^2$ in their entropy loss is also important. Without softmax normalization, $\ln \sigma$ inside cross-entropy $H(y, \sigma)$ turns into a linear term w.r.t. logits $\mathbf{v}^\top x$ and adding $\|\mathbf{v}\|^2$ creates a quadratic form resembling squared errors $(x - \mathbf{v})^2$ in K-means. In contrast, Section 2.2 shows that regularization $\|\mathbf{v}\|^2$ corresponds to the *margin maximization* controlling the width of the soft gap between the clusters, see our Fig.1(b).

In general, Figure 1 highlights fundamental differences between generative and discriminative approaches to clustering using two basic linear methods of similar parametric complexity (about $K \times M$ parameters). $K$-means (a) seeks balanced compact clusters of the least variance (squared errors). This can be interpreted "generatively" Kearns et al. (1997) as MLE fitting of two (isotropic) Gaussian densities, which also explains why K-means fails on highly anisotropic clusters (a). To fix this "generatively", one should use non-isotropic Gaussian densities. In particular, 2-mode GMM

would produce soft clusters as in (b). But, this increases parametric complexity (two extra covariance matrices) and leads to quadratic decision boundaries. In contrast, discriminative entropy clustering in (b) simply looks for the best linear decision boundary giving balanced ("fair") clusters with data points away from the boundary ("decisiveness"), regardless of the data density model complexity.

## 2.2 Entropy-based clustering and SVM: margin maximization

This section discusses similarities between entropy clustering with soft-max models and unsupervised SVM methods Ben-Hur et al. (2001); Xu et al. (2004). First, consider the fully supervised setting, where the relationship between SVMs Vapnik (1995) and logistic regression is known. Assuming binary classification with target labels $t = \pm 1$, one standard soft-margin SVM loss formulation combines a margin-maximization term with the *hinge loss* penalizing margin violations, e.g. see Bishop (2006)

$$L_{svm} = \gamma\|\mathbf{v}\|^2 + \overline{\max\{0, 1 - t\,\mathbf{v}^\top f\}} \tag{7}$$

where the linear classifier norm $\|\mathbf{v}\|$ (excluding bias!) is the reciprocal of the decision margin and $\gamma$ is the relative weight of the margin maximization term. For shortness and consistently with the notation introduced in Sec.1.1, logits $\mathbf{v}^\top f$ include the bias using "homogeneous" representations of $\mathbf{v}$ and features $f$, and the "bar" operator represents averaging over all training data points.

Instead of the hinge loss, soft-margin maximization (7) can use the *logistic regression* as an alternative soft penalty for margin violations, see Section 7.1.2 and Figure 7.5 in Bishop (2006),

$$L_{log} = \gamma\|\mathbf{v}\|^2 + \overline{\ln\left(1 + \exp^{-t\mathbf{v}^\top f}\right)} \equiv \gamma\|\mathbf{v}\|^2 + \overline{H(y, \sigma)} \tag{8}$$

where the second *binary cross-entropy* formulation in (8) replaces integer targets $t \in \{\pm 1\}$ with one-hot target distributions $y \in \{(1, 0), (0, 1)\}$ consistent with our general terminology in Sec.1.2. Our second formulation in (8) uses soft-max $\sigma = \{\frac{\exp l_1}{\exp l_1 + \exp l_2}, \frac{\exp l_2}{\exp l_1 + \exp l_2}\}$ with logits $l_1 = \frac{1}{2}\mathbf{v}^\top f$ and $l_2 = -\frac{1}{2}\mathbf{v}^\top f$; its one advantage is a trivial multi-class generalization. The difference between the soft-margin maximization losses (7) and (8) is that the flat region of the hinge loss leads to a sparse set of *support vectors* for the maximum margin solution, see Section 7.1.2 in Bishop (2006).

Now, consider the standard SVM-based self-labeling formulation of maximum margin clustering by Xu et al. (2004). They combine loss (7) with a linear *fairness* constraint $-\epsilon \leq \bar{t} \leq \epsilon$

$$L_{mm} = \gamma\|\mathbf{v}\|^2 + \overline{\max\{0, 1 - t\,\mathbf{v}^\top f\}}, \quad \text{s.t.} \quad -\epsilon \leq \bar{t} \leq \epsilon \tag{9}$$

and treat labels $t$ as optimization variables in addition to model parameters. Note that the hinge loss encourages consistency between the pseudo labels $t \in \{\pm 1\}$ and the sign of the logits $\mathbf{v}^\top f$. Besides, loss (9) still encourages maximum margin between the clusters. Keeping data points away from the decision boundary is similar to the motivation for the *decisiveness* in entropy-based clustering.

It is easy to connect (9) to self-labeling entropy clustering. Similarly to (7) and (8), one can replace the hinge loss by cross-entropy as an alternative margin-violation penalty. As before, the main difference is that the margin may not be defined by a sparse subset of *support vectors*. We can also replace the linear balancing constraint in (9) by an entropy-based fairness term. Then, we get

$$L_{semm} = \gamma\|\mathbf{v}\|^2 + \overline{H(y, \sigma)} - H(\bar{y}) \tag{10}$$

which is a self-labeling surrogate for the entropy-based maximum-margin clustering loss

$$L_{emm} = \gamma\|\mathbf{v}\|^2 + \overline{H(\sigma)} - H(\bar{\sigma}). \tag{11}$$

Losses (11) and (10) are examples of general clustering losses for $K \geq 2$ combining decisiveness and fairness as in Sections 1.1, 1.2. The first term can be seen as a special case of the norm regularization in (4). However, instead of a generic model simplicity argument used to justify (4), the specific combination of cross-entropy with regularizer $\|\mathbf{v}\|^2$ (excluding bias) in (11) and (10) is explicitly linked to margin maximization where $\frac{1}{\|\mathbf{v}\|}$ corresponds to the margin's width[2].

It was known that "for a poorly regularized classifier" the combination of decisiveness and fairness "alone will not necessarily lead to good solutions to unsupervised classification" (Bridle et al. (1991)) and that decision boundary can tightly pass between the data points (Fig.1 in Krause et al. (2010)). The formal relation to margin maximization above complements such prior knowledge. Our supplementary material (A) shows the empirical effect of parameter $\gamma$ in (11) on the inter-cluster gaps.

---

[2]The entropy clustering loss (6) is also appended with regularization $\|\mathbf{v}\|^2$ in Jabi et al. (2021), where it is incorrectly used for proving K-means connection, see Sec.2.1. They do not discuss margin maximization.

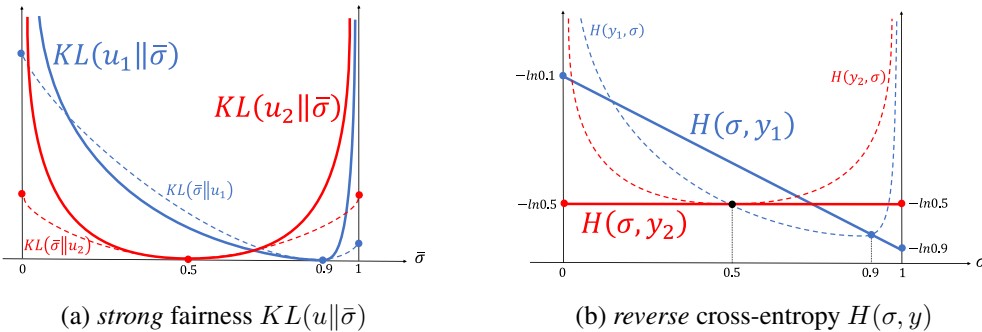

(a) strong fairness $KL(u\|\bar{\sigma})$      (b) reverse cross-entropy $H(\sigma, y)$

Figure 2: "Forward" vs "reverse": (a) KL-divergence and (b) cross-entropy. Assuming binary classification $K = 2$, probability distributions $\sigma$ or $\bar{\sigma}$ are represented as points on [0,1]. The solid curves in (a) illustrate the *forward* KL-divergence $KL(u\|\bar{\sigma})$ for average predictions $\bar{\sigma}$. We show two examples of volumetric prior $u_1 = (0.9, 0.1)$ (blue) and $u_2 = (0.5, 0.5)$ (red). The reverse KL-divergence $KL(\bar{\sigma}\|u)$ (dashed curves), commonly representing fairness in prior work, tolerates extremely unbalanced clustering, i.e. the end points of the interval [0,1]. The solid curves in (b) are the *reverse* cross-entropy $H(\sigma, y)$ for predictions $\sigma$. The dashed curves are the forward cross-entropy $H(y, \sigma)$. The plots in (b) show examples for two fixed pseudo-labels $y_1 = (0.9, 0.1)$ (blue) and $y_2 = (0.5, 0.5)$ (red). Our loss $H(\sigma, y)$ weakens the training (reduces gradients) on data points with higher label uncertainty (compare blue and red curves). In contrast, the standard loss $H(y, \sigma)$ trains the network to copy this uncertainty, see the optimum $\sigma$ on the dashed curves. The boundedness of $H(\sigma, y)$ also represents robustness to errors in $y$.

## 3    Our self-labeling entropy clustering method

The conceptual properties discussed in the previous section may improve the general understanding of entropy clustering, but their new practical benefits are limited. For example, margin maximization implicitly happens in prior entropy methods since norm regularization (weight-decay) is omnipresent.

This section addresses some specific limitations of prior entropy clustering formulations that do affect the practical performance. We focus on self-labeling (Sec.1.2) and observe that the standard cross-entropy formulation of decisiveness is sensitive to pseudo-label errors. Section 3.1 introduces our new self-labeling loss using the *reverse cross-entropy*, which we show is more robust to label noise. We also propose *strong fairness*. Section 3.2 derives an efficient EM algorithm for minimizing our loss w.r.t. pseudo-labels, which is a critical step of our self-labeling algorithm.

### 3.1    Our self-labeling loss formulation

We start from the maximum-margin entropy clustering (10) where the entropy fairness can be replaced by an equivalent KL-divergence term explicitly expressing the target balance distribution $u$. This gives a self-labeling variant of the loss (4) in Krause et al. (2010) similar to (6) in Jabi et al. (2021)

$$L_{semm} \;\overset{c}{=}\; \overline{H(y, \sigma)} \;+\; KL(\bar{y}\,\|\,u) \;+\; \gamma\,\|\mathbf{v}\|^2. \tag{12}$$

We propose two changes to this loss based on several numerical insights leading to a significant performance improvement over Krause et al. (2010) and Jabi et al. (2021). First, we reverse the order of the cross-entropy arguments, see Fig.2(b). This improves the robustness of network predictions $\sigma$ to errors in estimated pseudo-labels $y$, as confirmed by our experiment in Figure 3. This reversal also works for estimating pseudo-labels $y$ as the second argument in cross-entropy is a standard position for an "estimated" distribution. Second, we also observe that the standard fairness term in (12,4,6) is the *reverse* KL divergence w.r.t. cluster volumes, i.e. the average predictions $\bar{\sigma}$. It can tolerate highly unbalanced solutions where $\bar{\sigma}_k = 0$ for some cluster $k$, see the dashed curves in Fig.2(a). We propose the *forward*, a.k.a. *zero-avoiding*, KL divergence $KL(u\,\|\,\bar{\sigma})$, see the solid curves Fig.2(a), which assigns infinite penalties to highly unbalanced clusters. We refer to this as *strong fairness*.

The two changes above modify the clustering loss (12) into our formulation of self-labeling loss

$$L_{our} \;:=\; \overline{H(\sigma, y)} \;+\; \lambda\, KL(u\,\|\,\bar{y}) \;+\; \gamma\,\|\mathbf{v}\|^2. \tag{13}$$

### 3.2 Our EM algorithm for pseudo-labels

Minimization of a self-supervised loss w.r.t pseudo-labels $y$ for given predictions $\sigma$ is a critical operation in iterative self-labeling techniques Asano et al. (2020); Jabi et al. (2021), see Sec.1.2. Besides well-motivated numerical properties of our new loss (13), in practice it also matters that it has an efficient solver for pseudo-labels. While (13) is convex w.r.t. $y$, optimization is done over a probability simplex and a good practical solver is not a given. Note that $H(\sigma, y)$ works as a *log barrier* for the constraint $y \in \Delta^K$. This could be problematic for the first-order methods, but a basic Newton's method is a good match, e.g. Kelley (1995). The overall convergence rate of such second-order methods is fast, but computing the Hessian's inverse is costly, see Table 1. Instead, we derive a more efficient *expectation-maximization* (EM) algorithm.

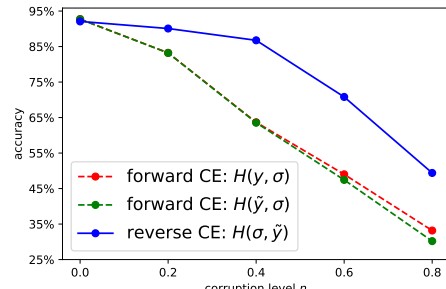

Figure 3: Robustness to noisy labels: reverse $H(\sigma, y)$ vs standard cross-entropy $H(y, \sigma)$. We train ResNet-18 on fully-supervised *Natural Scene* dataset [NSD] where we corrupted some labels. The horizontal axis shows the corruption level, i.e. percentage $\eta$ of training images where correct ground truth labels were replaced by a random label. We use soft target distributions $\tilde{y} = \eta * u + (1 - \eta) * y$ that is a mixture of one-hot distribution $y$ for the observed corrupt label and the uniform distribution $u$, as in Müller et al. (2019). The vertical axis shows the test accuracy. Reverse cross-entropy improves robustness to high labeling errors.

Assume that model parameters and predictions in (13) are fixed, *i.e.* $\mathbf{v}$ and $\sigma$. Following *variational inference* Bishop (2006), we introduce $K$ auxiliary latent variables, distributions $S^k \in \Delta^N$ representing normalized support of each cluster $k$ over $N$ data points. In contrast, $N$ distributions $y_i \in \Delta^K$ show support for each class at every point $X_i$. We refer to each vector $S^k$ as a *normalized cluster $k$*. Note that here we focus on individual data points and explicitly index them by $i \in \{1, \ldots, N\}$. Thus, we use $y_i \in \Delta^K$ and $\sigma_i \in \Delta^K$. Individual components of distribution $S^k \in \Delta^N$ corresponding to data point $X_i$ is denoted by scalar $S_i^k$.

First, we expand our loss (13) using our new latent variables $S^k \in \Delta^N$

$$
\begin{aligned}
L_{our} &\overset{c}{=} \overline{H(\sigma, y)} + \lambda H(u, \bar{y}) + \gamma \|\mathbf{v}\|^2 && (14)\\
&= \overline{H(\sigma, y)} - \lambda \sum_k u^k \ln \sum_i S_i^k \frac{y_i^k}{S_i^k N} + \gamma \|\mathbf{v}\|^2 \\
&\leq \overline{H(\sigma, y)} - \lambda \sum_k \sum_i u^k S_i^k \ln \frac{y_i^k}{S_i^k N} + \gamma \|\mathbf{v}\|^2 && (15)
\end{aligned}
$$

Due to the convexity of negative $\log$, we apply Jensen's inequality to derive an upper bound, i.e. (15), to $L_{our}$. Such a bound becomes tight when:

$$
\text{E-step}: \qquad S_i^k = \frac{y_i^k}{\sum_j y_j^k} \qquad\qquad (16)
$$

Then, we fix $S_i^k$ as (16) and solve the Lagrangian of (15) with simplex constraint to update $y$ as:

$$
\text{M-step}: \qquad y_i^k = \frac{\sigma_i^k + \lambda N u^k S_i^k}{1 + \lambda N \sum_c u^c S_i^c} \qquad\qquad (17)
$$

We run these two steps until convergence with respect to some predefined tolerance. Note that the minimum $y$ is guaranteed to be globally optimal since (14) is convex w.r.t. $y$. The empirical convergence rate is within 15 steps on MNIST. The comparison of computation speed on synthetic data is shown in Table 1. While the number of iterations to convergence is roughly the same as Newton's methods, our EM algorithm is much faster in terms of running time and is extremely easy to implement using the highly optimized built-in functions from the standard PyTorch library that supports GPU.

| | number of iterations (to convergence) | | | running time in sec. (to convergence) | | |
|---|---|---|---|---|---|---|
| **K** | 2 | 20 | 200 | 2 | 20 | 200 |
| Newton | 3 | 3 | 4 | $2.8e^{-2}$ | $3.3e^{-2}$ | $1.7e^{-1}$ |
| EM | 2 | 2 | 2 | $9.9e^{-4}$ | $2.0e^{-3}$ | $4.0e^{-3}$ |

Table 1: Our EM algorithm vs Newton's methods Kelley (1995).

Inspired by Springenberg (2015); Hu et al. (2017), we also adapted our EM algorithm to allow for updating $y$ within each batch. In fact, the mini-batch approximation of (14) is an upper bound. Considering the first two terms of (14), we can use Jensen's inequality to get:

$$\overline{H(\sigma, y)} + \lambda\, H(u, \bar{y}) \quad \leq \quad \mathbb{E}_B\big[\overline{H_B(\sigma, y)} + \lambda\, H(u, \bar{y}_B)\big] \tag{18}$$

where $B$ is the batch randomly sampled from the whole dataset. Now, we can apply our EM algorithm to update $y$ in each batch, which is even more efficient. Compared to other methods Ghasedi Dizaji et al. (2017); Asano et al. (2020); Jabi et al. (2021) which also use the auxiliary variable $y$, we can efficiently update $y$ on the fly while they only update once or just a few times per epoch due to the inefficiency to update $y$ for the whole dataset per iteration. Interestingly, we found that it is actually important to update $y$ on the fly, which makes convergence faster and improves the performance significantly (see supplementary material). We use this "batch version" EM throughout all the experiments. Our full algorithm for the loss (13) is summarized in supplementary material.

# 4 Experimental results

Our experiments start from clustering on fixed features to joint training with feature learning. We test our approach on standard benchmark datasets with different network architectures. We also provide the comparison of different losses under weakly-supervised settings (see supplementary material).

**Dataset**   For the clustering problem, we use four standard benchmark datasets: MNIST Lecun et al. (1998), CIFAR10/100 Torralba et al. (2008) and STL10 Coates et al. (2011). We follow Ji et al. (2019) to use the whole dataset for training and testing unless otherwise specified.

**Evaluation**   As for the evaluation on clustering, we set the number of clusters to the number of ground-truth category labels and adopt the standard method Kuhn (1955) by finding the best one-to-one mapping between clusters and labels.

## 4.1 Clustering with fixed features

We compare our method against the state-of-the-art methods using fixed deep features generated by pre-trained (ImageNet) ResNet-50 He et al. (2016). We use a one-layer linear classifier for all losses except for K-means. We set $\lambda$ in our loss to 100. We use stochastic gradient descent with learning rate 0.1 to optimize the loss for 10 epochs. The batch size was set to 250. The coefficients for the margin maximization terms are set to 0.001, 0.02, 0.009, and 0.02 for MNIST, CIFAR10, CIFAR100 and STL10 respectively. As stated in Section 2.2, such coefficient is important for the optimal decision boundary, especially when features are fixed. If we simultaneously learn the representation/feature and cluster the data, we observed that the results are less sensitive to such coefficient.

|  | STL10 | CIFAR10 | CIFAR100 (20) | MNIST |
|---|---|---|---|---|
| K-means | 85.20%(5.9) | 67.78%(4.6) | 42.99%(1.3) | 47.62%(2.1) |
| MI-GD Bridle et al. (1991); Krause et al. (2010) | 89.56%(6.4) | 72.32%(5.8) | 43.59%(1.1) | 52.92%(3.0) |
| SeLa Asano et al. (2020) | 90.33%(4.8) | 63.31%(3.7) | 40.74%(1.1) | 52.38%(5.2) |
| MI-ADM Jabi et al. (2021) | 81.28%(7.2) | 56.07%(5.5) | 36.70%(1.1) | 47.15%(3.7) |
| MI-ADM⋆ Jabi et al. (2021) | 88.64%(7.1) | 60.57%(3.3) | 41.2%(1.4) | 50.61%(1.3) |
| Our | **92.2%(6.2)** | **73.48%(6.2)** | **43.8%(1.1)** | **58.2%(3.1)** |

Table 2: Comparison of different methods using fixed features. The numbers are the average accuracy and the standard deviation over 6 trials. ⋆: our "batch version" implementation of their method.

## 4.2 Joint clustering and feature learning

In this section, we train a deep network to jointly learn the features and cluster the data. We test our method on both a small architecture (VGG4) and a large one (ResNet-18). The only extra standard technique we add here is the self-augmentation, following Hu et al. (2017); Ji et al. (2019); Asano et al. (2020). The experimental settings and more details are given in the supplementary material.

To train the VGG4, we use random initialization for network parameters. From Table 3, it can be seen that our approach consistently achieves the most competitive results in terms of accuracy

(ACC). Most of the methods we compared in our work (including our method) are general concepts applicable to single-stage end-to-end training. To be fair, we tested all of them on the same simple architecture. But, these general methods can be easily integrated into other more complex systems.

| | STL10 | CIFAR10 | CIFAR100 (20) | MNIST |
|---|---|---|---|---|
| MI-D[$\star$] Hu et al. (2017) | 25.28%(0.5) | 21.4%(0.5) | 14.39%(0.7) | 92.90%(6.3) |
| IIC[$\star$] Ji et al. (2019) | 24.12%(1.7) | 21.3%(1.4) | 12.58%(0.6) | 82.51%(2.3) |
| SeLa[§] Asano et al. (2020) | 23.99%(0.9) | 24.16%(1.5) | **15.34%(0.3)** | 52.86%(1.9) |
| MI-ADM[§] Jabi et al. (2021) | 17.37%(0.9) | 17.27%(0.6) | 11.02%(0.5) | 17.75%(1.3) |
| MI-ADM[$\star$,§] Jabi et al. (2021) | 23.37%(0.9) | 23.26%(0.6) | 14.02%(0.5) | 78.88%(3.3) |
| Our[$\star$,§] | **25.33%(1.4)** | **24.16%(0.8)** | 15.09%(0.5) | **93.58%(4.8)** |

Table 3: Quantitative results of accuracy for unsupervised clustering methods with VGG4. We only use the 20 coarse categories for CIFAR100. We reuse the code published by Ji et al. (2019); Asano et al. (2020); Hu et al. (2017) and implemented the optimization for loss of Jabi et al. (2021) according to the paper. $\star$: all variables are updated for each batch. §: loss formula has pseudo-label.

As for the training of ResNet-18, we found that random initialization does not work well when we only use self-augmentation. We may need more training tricks such as auxiliary over-clustering, multiple heads, and more augmentations Ji et al. (2019). In the mean time, the authors from Van Gansbeke et al. (2020) proposed a three-stage approach for the unsupervised classification and we found that the pre-trained weight from their first stage is beneficial to us. For a fair comparison, we followed their experimental settings and compared ours to their second-stage results. Note that they split the data into training and testing. We also report two additional evaluation metrics, i.e. NMI and ARI.

In Table 4, we show the results using their pretext-trained network (stage one) as initialization for our entropy clustering. We use only our clustering loss together with the self-augmentation (one augmentation per image this time) to reach higher numbers than SCAN, as shown in the table below.

| | CIFAR10 | | | CIFAR100 (20) | | | STL10 | | |
|---|---|---|---|---|---|---|---|---|---|
| | ACC | NMI | ARI | ACC | NMI | ARI | ACC | NMI | ARI |
| SCAN Van Gansbeke et al. (2020) | 81.8 (0.3) | 71.2 (0.4) | 66.5 (0.4) | 42.2 (3.0) | **44.1** (**1.0**) | 26.7 (1.3) | 75.5 (2.0) | 65.4 (1.2) | 59.0 (1.6) |
| Our | **83.09** (**0.2**) | **71.65** (**0.1**) | **68.05** (**0.1**) | **46.79** (**0.3**) | 43.27 (0.1) | **28.51** (**0.1**) | **77.67** (**0.1**) | **67.66** (**0.3**) | **61.26** (**0.4**) |

Table 4: Quantitative comparison using network ResNet-18.

# 5   Conclusions

Our paper proposed a new self-labeling algorithm for discriminative entropy clustering, but we also clarify several important conceptual properties of this general methodology. For example, we disproved a theoretical claim in a recent TPAMI paper stating the equivalence between variance clustering (K-means) and discriminative entropy-based clustering. We also demonstrate that standard formulations of entropy clustering losses may lead to narrow decision margins. Unlike prior work on discriminative entropy clustering, we show that classifier norm regularization is important for margin maximization.

We also discussed several limitations of the existing self-labeling formulations of entropy clustering and propose a new loss addressing such limitations. In particular, we replace the standard (forward) cross-entropy by the *reverse cross-entropy* that we show is significantly more robust to errors in estimated soft pseudo-labels. Our loss also uses a strong formulation of the fairness constraint motivated by a *zero-avoiding* version of KL divergence. Moreover, we designed an efficient EM algorithm minimizing our loss w.r.t. pseudo-labels; it is significantly faster than standard alternatives, e.g Newton's method. Our empirical results improved the state-of-the-art on many standard benchmarks for deep clustering.

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
