# Supplementary Material

## A  Margin maximization for entropy-based clustering

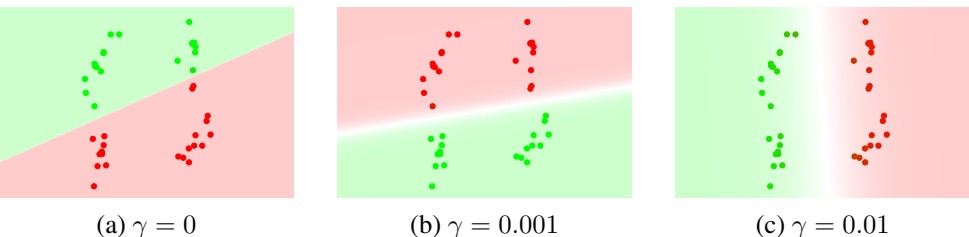

(a) $\gamma = 0$     (b) $\gamma = 0.001$     (c) $\gamma = 0.01$

Figure 4: Margin maximization term $\gamma \|\mathbf{v}\|^2$ in our loss (a): low-level clustering results for the softmax linear classifier model (3) with $N = 2$ and different weights $\gamma$. The dots represent data points. The optimal softmax clustering of the data and the decision regions over the whole space are visualized by $\sigma$-weighted color transparency, as in Fig.1(b). The "margin" is a weak-confidence "soft" region around the linear decision boundary lacking color-saturation. For small $\gamma$ the classifier can "squeeze" a narrow-margin linear decision boundary just between the data points, while maintaining arbitrarily hard "decisiveness" on the data points themselves.

The *average entropy* term in (1), a.k.a. "decisiveness", is recommended in Grandvalet & Bengio (2004) as a general regularization term for semi-supervised problems. They argue that it produces decision boundaries away from all training examples, labeled or not. This seems to suggest larger *classification margins*, which are good for generalization. However, the decisiveness may not automatically imply large margins if the norm of classifier $\mathbf{v}$ in posterior models (2, 3) is unrestricted, see Figure 4(a). Technically, this follows from the same arguments as in Xu et al. (2004) where regularization of the classifier norm is formally related to the margin maximization in the context of their SVM approach to clustering.

Interestingly, regularization of the norm for all network parameters $[\mathbf{v}, \mathbf{w}]$ is motivated in (4) differently Krause et al. (2010). But, since the classifier parameters $\mathbf{v}$ are included, coincidentally, it also leads to margin maximization. On the other hand, many MI-based methods Bridle et al. (1991); Ghasedi Dizaji et al. (2017); Asano et al. (2020) do not have regularizer $\|\mathbf{v}\|^2$ in their clustering loss, *e.g.* see (5). One may argue that practical implementations of these methods implicitly benefit from the *weight decay*, which is omnipresent in network training. It is also possible that gradient descent may implicitly restrict the classifier norm Soudry et al. (2018). In any case, since margin maximization is important for clustering, ideally, it should not be left to chance. Thus, the norm regularization term $\|\mathbf{v}\|^2$ should be explicitly present in any clustering loss for posterior models.

We extend MI loss (1) by combining it with the regularization of the classifier norm $\|\mathbf{v}\|$ encouraging *margin maximization*, as shown in Figure 4

$$
\begin{aligned}
L_{mi+mm} &:= \overline{H(\sigma)} - H(\overline{\sigma}) + \gamma \|\mathbf{v}\|^2 \\
&\overset{c}{=} \overline{H(\sigma)} + KL(\overline{\sigma} \,\|\, u) + \gamma \|\mathbf{v}\|^2.
\end{aligned} \tag{a}
$$

We note that Jabi et al. (2021) also extend their entropy-based loss (6) with the classifier regularization $\|\mathbf{v}\|^2$, but this extra term is used mainly as a technical tool in relating their loss (6) to K-means, as detailed in Section 2.1. They do not discuss its relation to margin maximization.

 # B  Proof

**Lemma B.1.** *Given fixed $\sigma_i \in \Delta^K$ where $i \in \{1, ..., M\}$ and $u \in \Delta^K$, the objective*

$$E(y) = -\frac{\beta}{M} \sum_i \sum_k \sigma_i^k \ln y_i^k - \lambda \sum_k u_k \ln \frac{\sum_i y_i^k}{M}$$

*is convex for $y$, where $y_i \in \Delta^K$.*

*Proof.* First, we rewrite $E(y)$

$$E(y) = -\sum_k \left( \frac{\beta}{M} \sum_i \sigma_i^k \ln y_i^k + \lambda u_k \ln \frac{\sum_i y_i^k}{M} \right)$$
$$:= -\sum_k f_k(y^k) \tag{b}$$

Next, we prove that $f_k : R_{(0,1)}^M \to R$ is concave based on the definition of concavityBoyd & Vandenberghe (2004) for any $k \in \{1, ..., K\}$. Considering $x = (1 - \alpha)x_1 + \alpha x_2$ where $x_1, x_2 \in R_{(0,1)}^M$ and $\alpha \in [0, 1]$, we have

$$
\begin{aligned}
f_k(x) = & \frac{\beta}{M} \sum_i \sigma_i^k \ln \left( (1 - \alpha)x_{1i} + \alpha x_{2i} \right) + \\
& \lambda u_k \ln \frac{\sum_i \left( (1 - \alpha)x_{1i} + \alpha x_{2i} \right)}{M} \\
\geq & \frac{\beta}{M} \sum_i (1 - \alpha)\sigma_i^k \ln x_{1i} + \alpha \sigma_i^k \ln x_{2i} \\
& + \lambda u_k \left( (1 - \alpha) \ln \frac{\sum_i x_{1i}}{M} + \alpha \ln \frac{\sum_i x_{2i}}{M} \right) \\
= & (1 - \alpha)f_k(x_1) + \alpha f_k(x_2)
\end{aligned}
$$

The inequality uses Jensen's inequality. Now that $f_k$ is proved to be concave, $-f_k$ will be convex. Then $E(y)$ can be easily proved to be convex using the definition of convexity with the similar steps above.

# C  Our Algorithm

---
**Algorithm 1** Optimization for our loss

---
**Input**  : network parameters $[\mathbf{v}, \mathbf{w}]$ and dataset
**Output** : network parameters $[\mathbf{v}^*, \mathbf{w}^*]$
**for** *each epoch* **do**
    **for** *each iteration* **do**
        Initialize $y$ by the network output at current stage as a warm start
        **while** *not convergent* **do**
            $S_i^k = \frac{y_i^k}{\sum_j y_j^k}$  $y_i^k = \frac{\sigma_i^k + \lambda N u^k S_i^k}{1 + \lambda N \sum_c u^c S_i^c}$
        **end**
        Update $[\mathbf{w}, \mathbf{v}]$ using loss $\overline{H_B(\sigma, y)} + \gamma \|\mathbf{v}\|^2$ via stochastic gradient descent
    **end**
**end**

---

## D   Loss Curve

In this section, we empirically show the faster convergence if we update $y$ for each batch after every iteration.

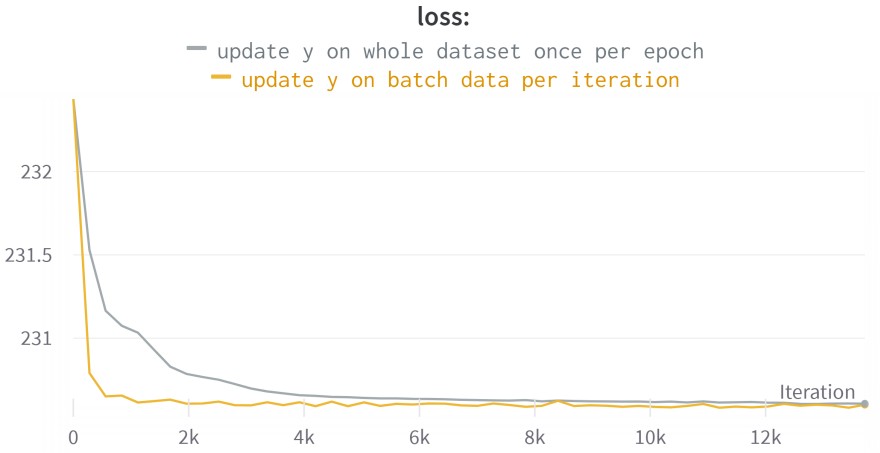

Figure 5: Loss (14) curves for different update setting on $y$. This is generated with just a linear classifier on MNIST. We use the same initialization and run both for 50 epochs. The gray line has an accuracy of 52.35% while the yellow one achieves 63%.

## E   Experiments

### E.1   Network Architecture

The network structure of VGG4 is adapted from Ji et al. (2019).

| Grey(28x28x1) | RGB(32x32x3) | RGB(96x96x3) |
|---|---|---|
| 1xConv(5x5,s=1,p=2)@64 | 1xConv(5x5,s=1,p=2)@32 | 1xConv(5x5,s=2,p=2)@128 |
| 1xMaxPool(2x2,s=2) | 1xMaxPool(2x2,s=2) | 1xMaxPool(2x2,s=2) |
| 1xConv(5x5,s=1,p=2)@128 | 1xConv(5x5,s=1,p=2)@64 | 1xConv(5x5,s=2,p=2)@256 |
| 1xMaxPool(2x2,s=2) | 1xMaxPool(2x2,s=2) | 1xMaxPool(2x2,s=2) |
| 1xConv(5x5,s=1,p=2)@256 | 1xConv(5x5,s=1,p=2)@128 | 1xConv(5x5,s=2,p=2)@512 |
| 1xMaxPool(2x2,s=2) | 1xMaxPool(2x2,s=2) | 1xMaxPool(2x2,s=2) |
| 1xConv(5x5,s=1,p=2)@512 | 1xConv(5x5,s=1,p=2)@256 | 1xConv(5x5,s=2,p=2)@1024 |
| 1xLinear(512x3x3,K) | 1xLinear(256x4x4,K) | 1xLinear(1024x1x1,K) |

Table 1: Network architecture summary. s: stride; p: padding; K: number of clusters. The first column is used on MNIST Lecun et al. (1998); the second one is used on CIFAR10/100 Torralba et al. (2008); the third one is used on STL10 Coates et al. (2011). Batch normalization is also applied after each Conv layer. ReLu is adopted for non-linear activation function.

We used standard ResNet-18 from PyTorch library as the backbone architecture for Figure 3. As for the ResNet-18 used for Table 4, we used the code from this repository [1].

### E.2   Ablation Study on Toy Example

We conducted an ablation study on toy examples as shown in Figure. 6. We use the normalized X-Y coordinates of the data points as the input. We can see that each part of our loss is necessary for obtaining a good result. Note that, in Figure 6 (a), (c) of 3-label case, the clusters formed are the

---

[1] `https://github.com/wvangansbeke/Unsupervised-Classification`

same, but the decision boundaries which implies the generalization are different. This emphasizes the importance of including $L2$ norm of $\mathbf{v}$ to enforce maximum margin for better generalization.

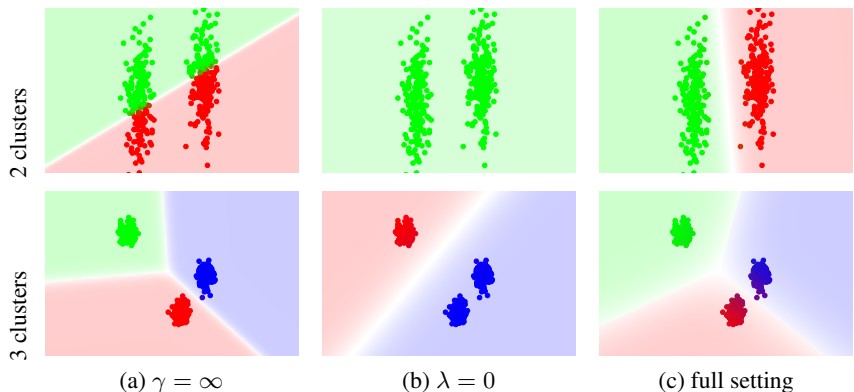

(a) $\gamma = \infty$        (b) $\lambda = 0$        (c) full setting

Figure 6: "Shallow" ablation study on toy examples.

### E.3 Deep Clustering

Here we present the missing experimental settings in Section 4.2. As for the training of VGG4, we use Adam Kingma & Ba (2015) with learning rate $1e^{-4}$ for optimizing the network parameters. We set batch size to 250 for CIFAR10, CIFAR100 and MNIST, and we use 160 for STL10. We achieved the self-augmentation by setting $\sigma_i = \mathbb{E}_t[\sigma(\mathbf{v}^\top f_\mathbf{w}(t(X_i)))]$. For each image, we generate two augmentations sampled from "horizontal flip", "rotation" and "color distortion". We set the $\lambda$ to 100 in our loss and use 1.3 as the weight of fairness term in (1). The weight decay coefficient is set to 0.01. We report the mean accuracy and Std from 6 runs with different initializations while we use the same initialization for all methods in each run. We use 50 epochs for each run and all methods reach convergence within 50 epochs.

As for the training of ResNet-18, we still use the Adam optimizer, and the learning rate is set to $1e^{-1}$ for the linear classifier and $1e^{-5}$ for the backbone. The weight decay coefficient is set to $1e^{-4}$. The batch size is 200 and the number of total epochs is 50. The $\lambda$ is still set to 100. We only use one augmentation per image, and we use an extra reverse cross-entropy loss to enforce the prediction of the augmentation to be close to the pseudo-label. The coefficient for such extra loss is set to 0.5, 0.2 and 0.4 respectively for STL10, CIFAR10 and CIFAR100 (20) datasets. We will release the training code.

### E.4 Weakly-supervised Classification

We additionally conducted experiments for weakly-supervised classification on STL10. We split the STL10 dataset into 5000 training images and 8000 testing images. We only keep a certain percentage of ground-truth labels for each class of training data. The accuracy is calculated on test set by comparing the hard-max of prediction to the ground-truth.

We use the same experimental settings as that in unsupervised clustering with VGG4 except for two points: 1. We add cross-entropy loss on labelled data; 2. We separate the training data from test data while we use all the data for training and test in unsupervised clustering. The results are shown in the following table.

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

|  | 0.1 | 0.05 | 0.01 |
|---|---|---|---|
| Only seeds | 40.27% | 36.26% | 26.1% |
| + MI-D Hu et al. (2017) | **47.39%** | 40.73% | 26.54% |
| + IIC Ji et al. (2019) | 44.73% | 33.6% | 26.17% |
| + SeLa Asano et al. (2020) | 44.84% | 36.4% | 25.08% |
| + MI-ADM Jabi et al. (2021) | 45.83% | 40.41% | 25.79% |
| + Our | 47.20% | **41.13%** | **26.76%** |

Table 2: Quantitative results for weakly-supervised classification on STL10. 0.1, 0.05 and 0.01 correspond to different ratios of labels used for supervision. "Only seeds" means that we only use standard cross-entropy loss on labeled training data.

Coates, A., Ng, A., and Lee, H. An analysis of single-layer networks in unsupervised feature learning. In *Proceedings of the fourteenth international conference on artificial intelligence and statistics*, pp. 215–223. JMLR Workshop and Conference Proceedings, 2011.

Ghasedi Dizaji, K., Herandi, A., Deng, C., Cai, W., and Huang, H. Deep clustering via joint convolutional autoencoder embedding and relative entropy minimization. In *Proceedings of the IEEE international conference on computer vision*, pp. 5736–5745, 2017.

Grandvalet, Y. and Bengio, Y. Semi-supervised learning by entropy minimization. *Advances in neural information processing systems*, 17, 2004.

Hu, W., Miyato, T., Tokui, S., Matsumoto, E., and Sugiyama, M. Learning discrete representations via information maximizing self-augmented training. In *International conference on machine learning*, pp. 1558–1567. PMLR, 2017.

Jabi, M., Pedersoli, M., Mitiche, A., and Ayed, I. B. Deep clustering: On the link between discriminative models and k-means. *IEEE Transactions on Pattern Analysis and Machine Intelligence*, 43 (6):1887–1896, 2021.

Ji, X., Henriques, J. F., and Vedaldi, A. Invariant information clustering for unsupervised image classification and segmentation. In *Proceedings of the IEEE/CVF International Conference on Computer Vision*, pp. 9865–9874, 2019.

Kingma, D. P. and Ba, J. Adam: A method for stochastic optimization. In *ICLR (Poster)*, 2015.

Krause, A., Perona, P., and Gomes, R. Discriminative clustering by regularized information maximization. *Advances in neural information processing systems*, 23, 2010.

Lecun, Y., Bottou, L., Bengio, Y., and Haffner, P. Gradient-based learning applied to document recognition. *Proceedings of the IEEE*, 86(11):2278–2324, 1998.

Soudry, D., Hoffer, E., Nacson, M. S., Gunasekar, S., and Srebro, N. The implicit bias of gradient descent on separable data. *The Journal of Machine Learning Research*, 19(1):2822–2878, 2018.

Torralba, A., Fergus, R., and Freeman, W. T. 80 million tiny images: A large data set for nonparametric object and scene recognition. *IEEE transactions on pattern analysis and machine intelligence*, 30 (11):1958–1970, 2008.

Xu, L., Neufeld, J., Larson, B., and Schuurmans, D. Maximum margin clustering. In Saul, L., Weiss, Y., and Bottou, L. (eds.), *Advances in Neural Information Processing Systems*, volume 17. MIT Press, 2004.