# OpenReview forum: "Discriminative Entropy Clustering and its Relation to K-means and SVM"
_NeurIPS.cc/2023/Conference — Submitted to NeurIPS 2023_

### Official Review · Reviewer_aVDq · 2023-06-20

**Soundness:** 2 fair
**Presentation:** 1 poor
**Contribution:** 2 fair
**Rating:** 4
**Confidence:** 4

**Summary:**

In this paper, the authors first presented an analysis of the relationship between the regularized information maximization (RIM) clustering framework to K-means and SVM-based clustering methods, showing stronger similarities to the SVM-based clustering than K-means. Then they proposed a new loss function and associated EM optimization algorithm for clustering leveraging the reverse cross-entropy/KL divergence to obtain more robust and fair clustering, which has been demonstrated to improve the performance on several balanced image classification benchmarks.

**Strengths:**

1. The authors identified an error about the missing normalization term in the proof of the equivalence theorem presented in Jabi et al. (2021).

2. The replacement of the forward cross-entropy with the reverse counterpart in the RIM loss appears novel in the clustering scenarios and has the potential to effectively mitigate the impact of uncertain/noisy pseudo-labels.

3. The proposed method showed improved performance on different image classification benchmarks over several baselines.

**Weaknesses:**

1. The manuscript was poorly written. The authors dedicated more than two pages to discuss the general background of the information maximization clustering framework and related methods. However, these discussions were confusing and largely limited the space for presenting the actual contributions of this work. Furthermore, many terms, including concepts like H and KL divergence, are not explicitly defined or explained, which may cause difficulties to understand the differences between the forward and reverse formulations. Additionally, the claimed conceptual and algorithmic contributions seem to be independent of each other. It is unclear if any of these conceptual clarifications contribute the discovery of the new loss function.

2. The disproof of the equivalence theorem in Jabi et al. (2021) is not convincing. While the authors pointed out an error in the original proof, this does not necessarily eliminate the possibility that the equivalence itself remains valid. Furthermore, this work focused on the standard K-means objective (including Figure 1), whereas Jabi et al. (2021) considered a soft and regularized K-means loss.

3. The authors' claim that the L2 regularization is linked to margin maximization seems questionable. [1] demonstrated that margin maximization is a property of the loss function itself rather than the regularization, which serves to control the model complexity. Indeed, certain combinations of loss function and regularization are not margin-maximizing.

4. The experimental validation is limited. A more comprehensive evaluation of the proposed modifications to the loss function would involve investigating the individual and combined effects of these changes on selected benchmarks and then comparing the results with multiple established baselines. It is still unclear how each modification contributes to the final improvement. Although the authors presented the impact of the reverse cross-entropy modification, they did so within the fully supervised setting rather than unsupervised scenarios. Furthermore, the authors only considered balanced datasets and tested the clustering with the ground truth number of clusters. A more diverse set of experimental conditions, including unbalanced datasets and varying numbers of clusters, would provide a more robust evaluation of the proposed method. Both NMI and ARI metrics used in Table 4 are capable of handling different number of clusters. Additionally, the architecture used in Section 4.2 is different from that used in the baseline methods. It would be preferable to standardize the experimental settings, including model architecture, to be able to directly compare with the results in the literature. Lastly, the inability of the proposed method to properly train a deep network-based clustering model is a concern as well. Most of the tricks should be independent of the loss function modifications, especially the reverse cross-entropy term, thus can be naturally integrated together.

[1] Rosset et al., 2003. Margin maximizing loss functions.

**Questions:**

1. Could the tolerance of "highly unbalanced solutions where $\bar{\sigma}_k = 0$ for some cluster k" be a good thing, especially when we over-cluster in practice?

2. How were the coefficients of the norm regularization and KL divergence determined in the proposed method and relevant baselines? Were they shared across all the methods? How could we select these coefficients in practice in an objective way since the performance is very sensitive to these values?

3. The K-means results reported in Table 2 for CIFAR10/100 do not match with those in Table 3 in the IMSAT paper, but both employed the pretrained ResNet-50 extracted features. This raises the question of the reliability of the reported quantitative results.

4. Why only used the 20 coarse categories for the CIFAR100 benchmark?

5. The authors mentioned that "if we simultaneously learn the representation/feature and cluster the data, we observed that the results are less sensitive to such coefficient." But I could not find these results in the main text or supplementary material.

6. Table 4 presents a comparison with SCAN but does not include any comparisons with the baseline methods featured in Table 3 following the same two-stage training.

**Limitations:**

There is no discussion of limitations or potential societal impact.

---

> ### Author Rebuttal · Authors · 2023-08-07
>
> **the first two pages (introduction) are confusing... and limit the space from the actual contributions...**
>
> It would greatly help if the reviewer could point out specific confusing points and redundancies. Adding the definitions of cross-entropy and KL divergence sounds good, thanks.
>
> **"conceptual and algorithmic contributions seem to be independent of each other"**
>
> Indeed, the conceptual and algorithmic contributions in Sections 2 and 3 are fairly independent on a technical level. But, they are related as follows. Section 2 established the general theoretical properties for entropy clustering solutions, and Section 3 develops a good algorithm for finding such solutions. For example, it would be sad if someone implements EM algorithm for GMM, but does not know or misunderstands the conceptual properties of the GMM problem and its MLE solution.
>
> **"The disproof of the equivalence theorem in [1] is not convincing..."**
>
> The reviewer seems to agree that the specific technical mistake we pointed out in the proof of the equivalence theorem in [1] is fundamental and can not be easily rectified. Now that their "proof'' is invalidated, there is no reason for that claim to be valid. For example, we did not see in [1] any intuition that led them to believe that regularized entropy clustering and soft K-means might be equivalent, which could have motivated them to try to prove this. In general, it is rare that two arbitrary losses are equivalent. Without valid proof, their claim should now be considered only as speculation or conjecture that is not even clearly motivated by anything we can see.
>
> Having said all that, we do want to prove to the community that their claim itself can not be true and this is why we are making our best effort to rectify your just criticism of our counterexample, see the first point in the general section of the rebuttal. We really appreciate your help with this and we do believe that the revised Figure 1 addresses your concern.
>
>  **"The authors' claim that the L2 regularization is linked to margin maximization seems questionable..."**
>
> The reference [2] convinced us that it is misleading to refer to the regularizer $||v||^2$ as a "margin-maximization'' term in Section 2.2, see lines 170 and 173. Indeed, the properties of the penalty function (hinge loss or cross-entropy) are also important for the margin maximization effect, as proven in [2]. On the other hand, without the regularization term, entropy-based clustering could be degenerate, as known since [3, 4], even though specific optimization algorithms may also invoke margin maximization bias in cross-entropy [5].
> Other parts of Section 2.2 use terms like the "margin maximization effect'' only in the context of the combined losses (7,8,9,10,11), which should be appropriate, but we will definitely correct lines 170 and 173. We will also cite [2], which provides a perfect technical argument on line 177 to motivate the switch from the hinge loss in (7) to the cross-entropy in (8).
>
> **"Although the authors presented the impact of the reverse cross-entropy modification, they did so within the fully supervised setting rather than unsupervised scenarios."**
>
> We compared our clustering results on standard benchmarks with the algorithm by [1], which uses standard cross-entropy and standard KL divergence.
>
> **" Add more diverse set of experiments"**
>
>  Our paper includes experiments on all the benchmarks that any prior entropy clustering works ever used.
>
> **concerns about training features from scratch**
>
> Our work is not focused on representation learning, however we do at least as well as prior entropy clustering methods. We are very interested in addressing this issue in our future work.
>
> **“How were the coefficients of the norm regularization and KL divergence determined in the proposed ...”**
>
> The coefficients of the norm regularization should theoretically be very small inducing maximum margin as we discussed in the general reply. However, mildly larger ones may be beneficial to the optimization. Thus, we still tuned such parameter by using simple grid search as well as for the coefficient of KL term. We tuned these hyperparameters separately for each method that uses them.
>
> **"The K-means results reported in Table 2 for CIFAR10/100 do not match..."**
>
> First, those numbers in IMSAT were obtained over 5 years ago and the pre-trained weights in the ResNet50 could possibly be changed over time. Second, we once used their official code but could not reproduce their own results.
>
> **"Table 4 presents a comparison with SCAN but does not include any comparisons with the baseline methods..."**
>
> We added the most important two baselines in the table below and we can observe a clearer improvement over these baselines.
>
> | | **CIFAR10** | **CIFAR10** | **CIFAR10**  | **CIFAR20** | **CIFAR20** | **CIFAR20** | **STL10** | **STL10** | **STL10**
> |:---| :----:  |    :----:      |    :----:      |    :----:      |    :----:      |    :----:      |    :----:      |    :----:      |    ---:
> || ACC      | NMI       | ARI   | ACC      | NMI       | ARI   | ACC      | NMI       | ARI   |
> |SCAN|81.3(0.3)|71.2(0.4)|66.5(0.4)|42.2(3.0)|**44.1(1.0)**|26.7(1.3)|75.5(2.0)|65.4(1.2)|59.0(1.6)|
> |MIADM| 74.76(0.3)  |  69.17(0.2) |62.51(0.2)  | 43.47(0.5)  |42.85(0.4)| 27.78(0.4)|67.84(0.2) |60.33(0.5)| 51.67(0.6)  |
> |IMSAT| 77.64(1.3)  | 71.05(0.4) |64.85(0.3)  |43.68(0.4) |42.92(0.2)|26.47(0.1)|70.23(2.0) | 62.22(1.2)  | 53.54(1.1)  |
> |our|**83.09(0.2)**|**71.65(0.1)**|**68.05(0.1)**|**46.79(0.3)**|43.27(0.1)|**28.51(0.1)**|**77.67(0.1)**|**67.66(0.3)**|**61.26(0.4)**|
>
> [1] Jabi et al. (2021)
> [2] Rosset et al. (2003)
> [3] Bridle et al. (1991)
> [4] Krause et al. (2010)
> [5] The implicit bias of gradient descent on separable data. Soudry et al. (2018)

---

> > ### Comment · Reviewer_aVDq · 2023-08-14
> >
> > Thanks for the response, which has addressed some of the concerns. However, there are still several remaining. First, all clustering results are obtained using the "ground truth" number of clusters, but we rarely have this information in practice. Thus, it would be much better to test the robustness of the proposed method with different number of clusters. Both NMI and ARI metrics are capable of handling different number of clusters. Second, the authors still did not provide any explanation regarding why only used the 20 coarse categories for the CIFAR100 benchmark. It is unclear if the proposed method only shows benefits with limited number of clusters.

---

> > > ### Author Response · Authors · 2023-08-16
> > >
> > > Thank you for taking the time to clarify these remaining concerns. We simply did not have space in the original rebuttal to answer all of your detailed comments. We are happy for the opportunity to address these here.
> > >
> > > We would like to divide your comment into two relatively independent parts:  (A) how does our method work when the number of clusters $K$ is not known, and (B) evaluating our method for different but **fixed** $K$.
> > >
> > > **(A) Estimating the unknown number of clusters $K$**
> > >
> > > In general, the assumption of the known number of clusters is significant and limits the practical usefulness of any clustering method based on this assumption. We are acutely aware of this in the context of some practical applications in our lab (e.g. clustering of DNA sequences). We are highly interested in addressing this question in our future work.
> > >
> > > As far as our submission is concerned, we do not make any general claims that we solved the clustering problem, which is fundamentally ill-posed [1]. Thus, some assumptions always have to be made (ideally, specific to the application).
> > >
> > > More specifically, our paper is focused on the general entropy clustering methodology [2-8]. All these and other prior works on entropy clustering assume that the number of clusters is fixed (even though some of them do experiments for different **fixed** $K$ on either simpler or different problems, see (B)). This assumption is very similar to the fundamental assumption in the basic *K-means* methodology, which is even reflected in its name. This is not to say that K-means cannot be generalized in a way where $K$ becomes an estimated variable, but this requires additional terms added to the loss (e.g. AIC or BIC information theoretic criteria). In fact, similar ideas can be explored in the context of entropy clustering, but, this is a substantial topic on its own. We do have plans to continue in this direction in our future work.
> > >
> > > We can more clearly state that fixed $K$ is an assumption made by us and all the previous works on entropy clustering (at least those we are aware of).
> > >
> > > **(B) Evaluating performance for different fixed $K$**
> > >
> > > Evaluating robustness to different (but fixed) $K$ would be interesting, but no prior entropy clustering papers compare NMI and ARI metrics using different $K$ on the data like STL10, CIFAR10, CIFAR100. All such prior works [2-8] typically do not vary $K$ on these datasets - they use them with NMI and ARI for one fixed $K$ (the same as in our experiments). We only see two exceptions: [3] shows some results with various $K$ but on much simpler datasets, while [7] vary $K$ on ImageNet - they are focused on representation learning and do not report clustering quality metrics like NMI or ARI. Also, we agree with your point about testing $K>20$ on CIFAR100, but we simply stuck to the experiments that we saw in prior work on entropy clustering.
> > >
> > > Moreover, we do not see any technical reason why our approach would differ from other entropy clustering methods in its robustness to $K$. We agree that entropy clustering methods (in general) should be evaluated for robustness to $K$, but it may be best to combine such evaluation with the proper study of point (A) above and present these in a separate dedicated paper.
> > >
> > > [1] An impossibility theorem for clustering, Kleinberg, NeurIPS 2002
> > > [2] Briddle et al. 1991
> > > [3] Krause et al. 2010
> > > [4] Hu et al. 2017
> > > [5] Ghasedi Dizaji et al. 2017
> > > [6] Ji et al. 2019
> > > [7] Asano et al. 2020
> > > [8] Jabi et al. 2021

---

### Official Review · Reviewer_pLFZ · 2023-06-23

**Soundness:** 3 good
**Presentation:** 2 fair
**Contribution:** 3 good
**Rating:** 5
**Confidence:** 3

**Summary:**

This paper first discusses a number of general properties of  entropy clustering methods, including their relation to K-means and unsupervised SVM-based techniques.
Then the aurthors find that cross-entropy is not robust to pseudo-label errors in clustering.
Finally, this paper proposes a new loss function based on reverse KL divergence for clustering  to obtain more robust and fair clustering.

**Strengths:**

(1) The proposed  loss function is interesting and and seems to be effective to obtain more robust and fair clustering.

(2) The authors try to establish connections between entropy clustering methods and classical methods.


**Weaknesses:**

(1) This paper is not well organized. There are too many details for the proposed method. Some of them can be moved to Appendix.

(2) There can be more descriptions and examples about the proposed method.

(3) The proposed method is merely tested on three image datasets.

**Questions:**

(1) Will the authors open-source the code?

**Limitations:**

I mentioned them in the Weakness section.

---

> ### Author Rebuttal · Authors · 2023-08-10
>
> **"This paper is not well organized. There are too many details for the proposed method. Some of them can be moved to Appendix."**
>
> If possible, please indicate which specific parts of Section 3 you prefer to be in the supplementary materials.
>
> **"There can be more descriptions and examples about the proposed method."**
>
> We extended our previous Figure 1 illustrating the general properties of entropy clustering losses and added a new Figure 2 further illustrating the dependence on the regularization parameter $\gamma$ (both figures are in the rebuttal's PDF). The empirical example in Figure 3 (in the paper) motivates our reverse formulation of cross entropy as more robust to errors in the pseudo labels. Figure 2 (in the paper) conceptually motivates our stronger formulation of the fairness constraint. As far as our closed-form EM algorithm is concerned, its properties are similar to other known EM techniques. One good example is the famous EM algorithm used for estimating Gaussian Mixtures (GMMs).
>
> **"The proposed method is merely tested on three image datasets."**
>
> We have results for four datasets: MNIST, STL10, CIFAR10 and CIFAR100 (20). These are all the deep clustering benchmarks used in the previous papers on entropy clustering. Our method consistently achieves the best performance.
>
> **"Will the authors open-source the code?"**
>
> Yes, once the paper is accepted.

---

### Official Review · Reviewer_UHK9 · 2023-07-05

**Soundness:** 3 good
**Presentation:** 4 excellent
**Contribution:** 4 excellent
**Rating:** 7
**Confidence:** 4

**Summary:**

The paper presents a very interesting analysis of discriminative entropy clustering in the literature and their use for self-labeling highlighting clear interpretation of the conditional and marginal entropy terms as decisiveness (push to have confident predictions) and fairness (to encourage desired proportions in clusters). The paper analyzes the variants of this kind of losses and their connections. They also discusses the relationship of this loss to K-means and refines the previously mistaken connection in a previous paper to point out the distinct difference. They further point out the effectiveness of reverse cross-entropy in case of uncertainty error and forward KL term to make them more effective  in the endpoints of softmax  interval. They also propose to use the regularization of classifier weights for margin maximization similar to SVM. A closed form update is derived to compute the pseudolabels from the combined weighted loss and shown its efficacy in clustering experiments.

**Strengths:**

- I think the paper discusses beautifully the intricacies and usage of Mutual information based entropy clustering loss, which is widely used in self labeling/self supervised/ weak supervised learning. The flow of the discussion is to the point and tries to give interpretation of each terms in the loss in a concise manner by drawing connection among the variants.

- The paper discusses the the previous link of entropy clustering to K-means and identify the distinct difference to k-means which was in fault in previous literature. On top of that they find the usefulness of the classifier weight regularization used in previous proof to link to the loss explicitly for margin maximization similar to SVM based clustering.

- The use of reverse cross-entropy and forward KL term and the motivation behind it is explained  very well with the aid of Figure 2, so that they are more robust in case of uncertainty around the corner of the softmax simplex.

- The formulation of the EM algorithm is nice to make it work faster along with batchwise operation, showing its global optimal solution guarantee at convergence due to drawing convexity with formulated tight upper bound.
- The experimental results shows clear improvement according to their claim.

**Weaknesses:**


1. I would say the results of joint clustering and feature learning in Table 3 is not encouraging even when showing a very shallow network of VGG4, the improvement is not significant apart from  MNIST. But in Resnet-18,  the inductive bias learned from pretraining is helping, then the improvement from the proposed loss might not improve very significantly with the proposed loss.  Also in Table 4, the regularization on the feature extractor $\textbf{w}$ done or not in the loss or by weight decay?

2. The experiments on semi supervised learning could also be shown to understand more when some supervision is available. How the idea of using reverse cross-entropy could be used in case of labeled one-hot y in equation 13? or it will be the regular cross-entropy for labeled set and reverse for pseudolabels with updated y_i?

3. What if the loss in 13 is directly optimized with gradient descent instead of using the EM? Although, it seems if $y_i= \sigma_i$ the reverse cross-entropy does not change anything if not updating y_i with closed-form update, is it?


**Questions:**

- In Algorithm 1, after updating the pseudolabels y_i in closed form solutions, the loss in the gradient descent is without the KL term in ALgorithm 1 in appendix. Is it a typo or the KL term is not required here?

**Limitations:**

The limitations are not discussed.

---

> ### Author Rebuttal · Authors · 2023-08-10
>
> **"But in Resnet-18, the inductive bias learned from pretraining is helping, then the improvement from the proposed loss might not improve very significantly with the proposed loss. Also in Table 4, the regularization on the feature extractor $w$
>  done or not in the loss or by weight decay?"**
>
> We extended Table 4 to include previous (competing) entropy clustering methods. The revised Table can be found in the answers to reviewer aVDq. Our approach works significantly better than them. Please also note that all methods (including ours) use weight decay for $w$ (representation layers).
>
> **"The experiments on semi supervised learning could also be shown to understand more when some supervision is available. How the idea of using reverse cross-entropy could be used in case of labeled one-hot y in equation 13? or it will be the regular cross-entropy for labeled set and reverse for pseudolabels with updated $y_i$?"**
>
> We have some semi-supervised experiments in the supplementary materials, though not very extensive as we keep exploring the applications. Indeed, the reverse cross-entropy is not well-defined for one-hot $y$, but simple $\epsilon$-softening works fine in practice. Our semi-supervised experiments use standard cross-entropy for points with ground truth labels, but we know that it works the same with the reverse cross-entropy and $\epsilon$-softening.
>
> **"What if the loss in 13 is directly optimized with gradient descent instead of using the EM? Although, it seems if $y_i==\sigma_i$
>  the reverse cross-entropy does not change anything if not updating $y_i$ with closed-form update, is it?"**
>
> In fact, each $y$-update (the EM part of the overall algorithm) is always initialized from the current predictions, i.e. $y_i = \sigma_i$. Indeed, if using gradient descent for updating $y$, the decisiveness term based on our reverse cross entropy would give zero gradients.
> But, the fairness part of the loss will not, unless the current prediction are fair. In fact, EM algorithm would not change $y$ in this case as well (if predictions are fair). However, the update of prediction will work for our reverse cross-entropy even if the current predictions agree
> with the pseudo-labels $y_i = \sigma_i$, see the solid lines in Figure 2b, unless the predictions are uniform. The update will encourage certainty for predictions.
>
> **"In Algorithm 1, after updating the pseudolabels $y_i$ in closed form solutions, the loss in the gradient descent is without the KL term in ALgorithm 1 in appendix. Is it a typo or the KL term is not required here?"**
>
> This is not a typo. The fairness constraint is imposed for pseudo-labels only, which is common for self-labeling entropy clustering methods we review in our paper [1, 2]. It is also true for SVM clustering in [3]. This is actually intended as the difficult task of optimizing a challenging non-convex loss (combining decisiveness and fairness) is delegated to a dedicated specialized solver (e.g. EM algorithm in our case) rather than to the gradient descent (backpropagation).
>
> [1]  Asano et al. (2020)
> [2] Jabi et al. (2021)
> [3] Xu et al. (2004)

---

### Official Review · Reviewer_ed2e · 2023-07-06

**Soundness:** 3 good
**Presentation:** 2 fair
**Contribution:** 2 fair
**Rating:** 3
**Confidence:** 4

**Summary:**

The authors consider discriminative entropy clustering and produce a discussions linking previous works. They have a version of the algorithm based on EM and a modified KL-divergence term. Experiments show the modified algorithm works better than competing methods with small networks.


**Strengths:**

- The authors provide a good overview of discriminative entropy clustering and its development, from the work of Bridle et al to the regularized version by Krause et al, to the more recent work using deep learning and representation learning (Asano et al and Jabi et al).


**Weaknesses:**

- The contribution of this paper is somewhat limited:
  1. The pointing out of a proof error in Jabi et al is helpful but is not significant on its own
  2. The discussion on SVM is based on previous works and simply replaces the logistic loss with margin loss, and is not particularly insightful
  3. Section 3 is the authors' contributed new algorithm, but the main difference with previous works is changing the order of the KL term in the objective.

- The improvements in empirical evaluations, compared to other methods, are somewhat limited. Many of the improvements are within standard error of competing methods.

- The authors use a lot of space to discuss previous work (first 5.5 pages), and not enough space to explain what is new about their method and specifically what problems it addresses.


**Questions:**

- Is there any explanation on the joint clustering and feature learning working only on small networks like VGG4, but not on larger VGG or ResNet18?


**Limitations:**

Limitations not mentioned; potential negative societal impact not applicable.

---

> ### Author Rebuttal · Authors · 2023-08-10
>
> **"The pointing out of a proof error in [1] is helpful but is not significant on its own."**
>
> While we agree, there are many other contributions in our paper. For example, besides finding a fundamental problem in
> their proof, we also show a counterexample proving that their actual claim is wrong.
>
>
> **"The discussion on SVM is based on previous works and simply replaces the logistic loss with margin loss, and is not particularly insightful."**
>
> The technical contribution of Sec 2.2 is that it shows how to combine the arguments in [2, 3, 4] to demonstrate that regularized
> entropy clustering losses like (10) or (11) have (soft) margin maximization properties.
> The perceived simplicity of our arguments is compensated by the conceptual significance of the newly
> discovered property that was previously unknown for the entropy-based clustering introduced in [5].
>
>
> **"Section 3 is the authors' contributed new algorithm, but the main difference with previous works is changing the order of the KL term in the objective.''**
>
> Our EM algorithm addresses a new entropy clustering loss reversing the order of both cross-entropy (decisiveness) and KL divergence (fairness). We believe that both are well-motivated conceptually by robustness to (pseudo) label errors (see Fig.2 \& 3) and stronger fairness (Fig.2). Our new loss is also motivated empirically as the algorithm improves over the previous entropy clustering formulations.
>
> **"The improvements in empirical evaluations, compared to other methods, are somewhat limited. Many of the improvements are within the standard error of competing methods."**
>
> We respectfully disagree. First, the improvements are very consistent across all experiments and data sets.
> Sometimes they are small, but in some cases they are significant. We also added an extended version of Table 4 (included in our answer to reviewer aVDq). This table compares with the state-of-the-art deep clustering (not necessarily) entropy-based.
> This might be the most relevant experiment for practically-minded readers interested in clustering as this experiment
> removes the self-imposed constraints of other experiments (e.g. fixed-features or training features from scratch).
> The revised Table 4 shows fairly significant improvements, particularly over previous entropy clustering methods.
>
> **"The authors use a lot of space to discuss previous work (first 5.5 pages), and not enough space to explain what is new about their method and specifically what problems it addresses."**
>
> Please note that we consider Section 2 (pages 4-6) as a conceptual contribution of our paper studying entropy clustering,
> which we like a lot. Section 2 improves theoretical understanding of the properties of entropy clustering that are not well understood (e.g. margin maximization and relation to SVM) or even misunderstood in the ML community (false connection to K-means in a recent TPAMI paper). Section 3 also provides an algorithmic contribution, also based on an improved understanding of the limitation in prior formulations.
>
> **"Is there any explanation on the joint clustering and feature learning working only on small networks like VGG4, but not on larger VGG or ResNet18?"**
>
> Intuitively, larger networks are more difficult to train to obtain good features. In other words, there will be more local minima. From our experiments, good initialization helps to avoid bad local minima, see the revised Table 4.
>
> [1] Jabi et al. (2021)
> [2] Margin maximizing loss functions. Rosset et al., 2003
> [3] Bishop (2006)
> [4] Xu et.al. (2004)
> [5] Briddle et.al. (1991)

---

> > ### Comment · Reviewer_ed2e · 2023-08-18
> >
> > I would like to thank the authors for the rebuttal. I am still skeptical about the claimed contributions in Section 2.2 after the rebuttal. In classification it is common to switch between the hinge loss in SVM and cross entropy loss in logistic regression, and the linear class balance constraint used in Xu et al 2004 and the entropy regularization term are also well-known in semi-supervised learning literature. So what is the main new understanding in this section?
> >
> > As for the point on Section 2 (page 4-6), it still reads much more like a literature review than a description of new theoretical understanding. If there are new theoretical understanding, the authors could have stated it formally using a simple lemma or theorem. This makes it easier for reviewers to evaluate whether the claim is novel or significant.
> >
> > Updated Table 4 shows improved results, but the results in Tables 2 and 3 are mostly still within standard errors of top 2/3 competing methods. The issue with the empirical evaluations is not just whether the method beats the state-of-art, but whether the improvements come from the algorithmic advances the authors propose. Since the empirical difference is relatively small, how can we be sure the improvements come from the self-labeling mechanism or changes to the KL-term?

---

> > > ### Author Response · Authors · 2023-08-19
> > >
> > > >**"In classification it is common to switch between the hinge loss in SVM and cross entropy loss in logistic regression, ...  what is the main new understanding in Section 2?"**
> > >
> > > Indeed, "switching" between hinge loss and cross-entropy is known in SVM **classification**, as discussed in textbooks [1] that we cite. But we would like to emphasize that our paper is on **clustering**, which is a significantly different problem, in particular, it is **unsupervised**.
> > >
> > > We are first to point out the relation between entropy clustering and margin maximization, to the best of our knowledge. Perhaps convincing arguments in our paper make this obvious, but it was not known to many highly respected scientists working on entropy clustering [2-5].  These papers have no references to SVM clustering [6], maybe because it was not obvious to them that there is any relation. Moreover, the result published in 2021 in the top ML journal [5] claims that the entropy clustering is equivalent to K-means. This directly contradicts the relation to SVM clustering [6] since K-means has no margin maximization property.
> > >
> > > In short (and directly answering the reviewer's question), the main new understanding that our paper provides for the ML community is that entropy clustering has margin maximization property and closely relates to SVM-based clustering [6]. Instead, the community currently believes that entropy clustering is related to K-means, which is incorrect.
> > >
> > > >**"Section 2 still reads much more like a literature review than a description of new theoretical understanding. If there are new theoretical understanding, the authors could have stated it formally using a simple lemma or theorem."**
> > >
> > > Concerning Section 2.1, we do not see how an error that we found in the proof of the main result in [5] (equivalence to K-means) can be stated as a theorem. Neither do we understand how our counterexample to their claim (see Fig 1) can be stated as a theorem.  Does the reviewer have specific suggestions?
> > >
> > > Concerning Section 2.2, it might be possible to state some formal property on the relation between regularized entropy clustering and margin maximization (e.g. see an informal claim in the first part of the general rebuttal). However, a similar formal claim may be formalistic and weak as an assumption restricting the claim to fair solutions is unavoidable. A similar issue also exists in standard **soft** SVM formulations for classification. Such losses seek a compromise between margin size and margin violations, but it is difficult to formally define what max margin even means for non-separable labeled data. In our unsupervised setting, data doesn't even have ground truth and our combined loss for unsupervised clustering also includes a soft fairness constraint (e.g. KL divergence term). For now, we prefer not to make any formal claims in Sec 2.2 and leave this development for future work.
> > >
> > > We believe that Section 2.2 works sufficiently well without a formal claim. For example, [6] introduced "max-margin clustering" (as stated in their title) but they do not have any theorem or lemma proving that their method produces max-margin clusters, which are not even formally defined. Yet, most readers of that paper will probably be convinced (just like us) that their formulation of clustering is related to margin maximization. Their self-labeling methodology integrates the standard soft margin SVM formulation with a soft fairness constraint.
> > >
> > > **References**
> > >
> > > [1] Bishop (2006)
> > >
> > > [2] Briddle et.al. (NeurIPS 1991)
> > >
> > > [3] Krause et al. (NeurIPS 2010)
> > >
> > > [4] Hu et al. (ICML 2017)
> > >
> > > [5] Jabi et al. (TPAMI 2021)
> > >
> > > [6] Xu et.al. (NeurIPS 2004)

---

> > > > ### Author Response · Authors · 2023-08-19
> > > >
> > > > >**"Updated Table 4 shows improved results, but Tables 2,3 are mostly within standard errors..."**
> > > >
> > > > We are glad that the reviewer seems satisfied with our results in Table 4, which is important for practical clustering since it removes artificial constraints (fixed features or randomly initialized features without pretraining). Still, we agree that Tables 2 and 3 are highly relevant.
> > > >
> > > > Please note that the **consistency** of our method over different datasets/applications in Tables 2 and 3 is a significant advantage. Our method is the top performer on almost all benchmarks (including Tables 2 and 3), while most prior methods that come close on some of the examples are not consistent on others. Also, it is reassuring that we achieve consistent top performance using a mathematically transparent formulation.
> > > >
> > > > >**"how can we be sure the improvements come from the self-labeling mechanism or changes to the KL-term?"**
> > > >
> > > > Comparisons with the basic (non-self-labeling) formulation of entropy clustering [2,3] are present in Table 2 (MI-GD), Table 3 (MI-D), and Table 4 (IMSAT). The latter two include self-augmentation, which is why we cite other papers there. We will clarify this and perhaps unify the labels in these Tables. Interestingly, some prior self-labeling methods do not compare with these early works (for no clear reason) while performing worse, at least in our tests. To the best of our knowledge, our experiments are the most comprehensive empirical evaluation of the entropy clustering methods, which is one of the useful contributions of our work.
> > > >
> > > > Advantages due to the proposed changes in KL and cross-entropy terms should be clear from comparison with the entropy clustering formulation in [5] (see Tables 2,3,4) where the only difference is the order of variables in KL and cross-entropy terms. We can not easily test each of these "switches" independently as we could not formulate an efficient self-labeling algorithm when only one of the terms is switched. Note that the differences with [5] in Tables 2,3,4 are substantial, in some cases above 14%.
> > > >
> > > > Also, we directly compared the forward and reverse cross-entropy in the fully supervised setting demonstrating a significant advantage of our formulation (see Fig 3). We find this particular test significant, and it was also positively noted by other reviewers.
> > > >
> > > > **References**
> > > >
> > > > [1] Bishop (2006)
> > > >
> > > > [2] Briddle et.al. (NeurIPS 1991)
> > > >
> > > > [3] Krause et al. (NeurIPS 2010)
> > > >
> > > > [4] Hu et al. (ICML 2017)
> > > >
> > > > [5] Jabi et al. (TPAMI 2021)
> > > >
> > > > [6] Xu et.al. (NeurIPS 2004)

---

### Author Rebuttal · Authors · 2023-08-07

# General points for all reviewers
## Entropy Clustering and Margin Maximization (Sec 2.2)

Reference [1] provided by aVDq not only strengthens the arguments in Sec. 2.2 but also inspires some additional analysis that improves our understanding of how the regularization parameter $\gamma$ in entropy clustering loss (10) affects the maximum margin. While we address the specific concerns by aVDq in his reviewer's section, here we explain some extra insights on $\gamma$ in (10). We can incorporate such insights and improved illustrations into the final paper or the supplementary materials.

But first, we show the right place for reference [1] in Sec 2.2. On line 177 it works better than [2] (Sec. 7.1.2) to justify loss (8) as an alternative formulation for soft SVM classification (7). Note that the rest of Sec.2.2 extends margin maximization property to non-convex **entropy-based clustering** (10,11), while the theories in [1] are limited to convex **classification** problems assuming fully labeled separable data.

**Improved insights on $\gamma$**: Our submission treats $\gamma$ as a trade-off parameter in entropy clustering, e.g. (10,11). Such *margin-width-vs-margin-violation* interpretation is common for the soft SVM classification losses (7,8) used for non-separable labeled data. However, we overlooked the importance of the relation between such soft SVM and the classical SVM where the maximum margin is formally defined assuming linearly separable labeled data. As well known (e.g. Sec.7.1.1, [2]), assuming separability the soft margin solution in (7) or (8) converges to the classical max-margin solution as $\gamma\rightarrow 0$. That is, for separable data soft SVM produces the max-margin solution for all sufficiently small $\gamma>0$. We can extend this standard property to self-labeling clustering (9) or (10). Indeed, for a fixed pseudo-labeling, the optimal classifier $v$ should produce a max-margin solution for any given sufficiently small $\gamma>0$. The hinge or decisiveness terms in (9) or (10) should approach zero for any such solution where all data points respect the margin. Thus, among all balanced linearly separable pseudo-labelings, losses (9,10) should prefer one with the largest margin width $\frac{1}{||v||}$ corresponding to the lowest value of the regularization term $\gamma||v||^2$. Therefore, **assuming fairness, losses (9) and (10) prefer the maximum margin clustering for any given sufficiently small $\gamma>0$**.

**Improved figures**: The property above is illustrated for entropy clustering in the revised Figure 1 and new Figure 2 (in rebuttal's PDF) that focus on smaller $\gamma$. Besides decision boundaries of the optimal linear classifier $\sigma(v^\top f_p)$, our figures use transparency to visualize the softness of decisions produced by the soft-max $\sigma(\cdot)$, which appears as a boundary ''blur'' proportional to $\frac{1}{||v||}$. Max-margin solutions are also known to have width $\frac{1}{||v||}$. Thus, the boundary "blur'' of the optimal entropy clustering is proportional to the width of the corresponding max margin.

## Regularized Entropy Clustering vs. soft K-means (Sec 2.1)

Reviewer aVDq points out that our counterexample in Fig.1 uses hard K-means, which is not sufficient to contradict the claim in [3] about the equivalence between the soft K-means (sKM) and regularized entropy clustering (EC) as in (10). Please note that hard K-means in Fig 1 was used mainly for shortness, but we agree that this led to ambiguity. Here we hope to convince all reviewers
that in our counterexample sKM is not fundamentally different from the hard K-means, in part because we focus on small $\gamma$ revealing (as discussed above) the max-margin property of EC, which sKM does not have for any $\gamma$. Our modified version of Fig.1 and some extra examples in the new Fig.2 (see both in the rebuttal's PDF) illustrate significant differences between EC and sKM discussed below. They strengthen our counterexample to the equivalence claim in [3].

**Different global minima**: EC and sKM prefer fundamentally different solutions in our counterexample, compare (A) and (D) in the modified Fig.1. Each method produces consistent solutions for all $\gamma\in(0,0.00001]$ where the optimal loss varies below four significant digits. sKM is nearly identical to the hard K-means for this range of $\gamma$. In fact, the optimal solution for sKM is similar to (D) for all $\gamma\geq 0$, only the boundary softens for larger $\gamma$, see new Fig.2. Indeed, sKM is still a variance clustering criterion, even though the variance of each cluster is "weighted" based on each point's cluster membership that softens mainly near the boundary. Similarly to hard K-means, sKM prefers more compact clusters.

**Different local minima**: New Fig.1 shows the difference between local minima for EC and sKM. The local minimum in (C) is obtained by the standard sKM algorithm initialized by solution (A). Vice versa, if EC algorithm (e.g. [3]) is initialized by (C), it converges to (A). The same relationship applies to the two solutions in (B) and (D). Local minima for EC in (A) and (B) find balanced clustering with (locally) maximum margin, while local minima for sKM in (C) and (D) are always orthogonal bisectors for the cluster centers that ignore cluster margins.

**Different dependence on the parameter $\gamma$**: Note that $\gamma$ works as a temperature parameter in sKM, which quickly converges to hard clusters as $\gamma\rightarrow 0$, see new Figs.1 and 2. On the other hand, EC converges to the max-margin solution as $\gamma\rightarrow 0$ (see above). The boundary "blur'' represents the corresponding max-margin width $\frac{1}{||v||}$ (see the end of the previous section) explaining why the "blur'' of the optimal EC solutions in our figures becomes fixed as $\gamma\rightarrow 0$.

## References
[1] Margin maximizing loss functions. Rosset et al., 2003.

[2] Bishop (2006)

[3] Jabi et al. (2021)

---

### Decision · Program_Chairs · 2023-09-21

**Decision:**

Reject

**Comment:**

The paper deals with the properties of entropy clustering methods, their relationships to K-means and unsupervised SVM-based techniques. A new EM-based algorithm with a reverse KL divergence is proposed. Experiments on classic datasets are provided.


The reviews have recognized the interestingness of the study and comparisons with other works and the proposed methodology.
Reviewer was very positive on the contribution.
The other reviewers were more skeptical and raised reservations on the clarity of the paper, the experiments and the significance of the novelty of the work.
During the rebuttal, the authors have provided different answers to the remarks of the reviewers. A new experimental evaluation was in particular provided with convincing results.

However, overall the weaknesses outweigh the strengths of the paper and the evaluation makes the paper below the acceptance bar.
I propose then rejection.

I encourage the authors to try to revise the paper with respect to the different feedbacks received.